# Research on Temperature Compensation of Optical Fiber MEMS Pressure Sensor Based on Conversion Method

Guozhen Yao [1,2,3,*], Yongqian Li [1,2,3], Qiufeng Shang [1] and Hanbai Fan [1]

1. Department of Electronic and Communication Engineering, North China Electric Power University, Baoding 071003, China
2. Hebei Key Laboratory of Power Internet of Things Technology, North China Electric Power University, Baoding 071003, China
3. Baoding Key Laboratory of Optical Fiber Sensing and Optical Communication Technology, North China Electric Power University, Baoding 071003, China

\* Correspondence: ygz@ncepu.edu.cn

**Abstract:** The characteristics of optical fiber MEMS pressure sensors are easily affected by temperature, so effective temperature compensation can improve the accuracy of the sensor. In this paper, the temperature characteristics of optical fiber MEMS pressure sensors are studied, and a temperature compensation method by converting the wavelength is proposed. The influence of target temperature and data point selection on the compensation effect is studied, and the effectiveness of the method is verified by the temperature compensation of sensors before and after aging. When the converted target temperature is 25 °C, the pressure measurement accuracy of the sensor is improved from 1.98% F.S. to 0.38% F.S. within the range of 5–45 and 0–4 MPa. The method proposed in this paper can not only improve the accuracy but also make the regular calibration more operable.

**Keywords:** conversion method; temperature compensation; optical fiber MEMS F-P pressure sensor; sensor calibration





## 1. Introduction

Due to the advantages of optical sensors, such as high sensitivity, fast response, and anti-electromagnetic interference, the use of optical sensing to detect physical quantities such as pressure, temperature, and the refractive index has received great attention. Researchers have designed a variety of optical sensors to measure pressure or temperature, such as Micro-Electro-Mechanical System (MEMS) structure [1–3], fiber Bragg grating [4–7], Mach-Zehnder interferometer [8,9] and microstructure [10–14].

Optical fiber MEMS pressure sensors have the characteristics of small volume, light weight, anti-electromagnetic interference, and tolerance to harsh environments [15] and are widely used in biomedicine, environmental monitoring, and other fields [16–18].

The main part of the sensor is Fabry–Perot cavity (F-P cavity). When the pressure acts on one end face of the F-P cavity, the cavity length will change, which further causes the wavelength of the F-P cavity to change to realize pressure sensing [19]. The F-P cavity is composed of silicon wafers [15,20–23]. When the external temperature changes, the thermal expansion and cold contraction effect of silicon wafers and the influence of residual pressure will lead to changes in the cavity length [22,24–27]. When the sensor is in an environment of large temperature change, if no measures are taken, it will not be able to distinguish whether the wavelength change is caused by temperature change or pressure change, resulting in pressure measurement error. Therefore, temperature compensation of MEMS pressure sensors is required to achieve high-accuracy pressure sensing. Moreover, because of the creep characteristics of MEMS sensors, it is necessary to calibrate the sensors after a period of use. Due to the influence of temperature on the pressure characteristics, the pressure-wavelength relationship must be calibrated at multiple temperature points

when calibrating the sensor, which makes the calibration process very cumbersome and the operability poor.

Aiming at the problem of the temperature influence of MEMS pressure sensors, some scholars analyzed or tested the temperature characteristics but did not give specific solutions or ignored the temperature effect. Jiang Xiaofeng et al. [28] designed an optical fiber F-P pressure sensor based on white light interference demodulation, and the temperature sensitivity was about 1.4 nm/°C. Wu Zhenhai et al. [29] conducted modeling and experimental analysis on the factors influencing the temperature performance of the MEMS pressure sensor and obtained the conclusion that the temperature sensitivity was 0.2333 nm/°C in the ideal model. Ming Li et al. [15] designed a simple and small optical fiber F-P pressure sensor using surface and bulk MEMS technology but did not give a solution to the temperature dependence. Xiaoguang Qi et al. [20] designed an optical fiber F-P pressure sensor with MEMS microcavity structure, which can realize ultra-high pressure detection, and the temperature effect was better than $2.665 \times 10^{-3}$ rad/°C. Fei Feng et al. [30] designed a MEMS pressure sensor suitable for working in high-temperature environments, and the results showed that the pressure sensitivity increased with temperature. Jia Liu et al. [31] designed an optical fiber F-P pressure sensor based on MgO single crystals for harsh environment monitoring and measured the change of sensor sensitivity and zero drift with temperature, which are 0.00131 μm/(MPa·°C) and 0.00353 μm/°C respectively.

In order to reduce the influence of temperature on the MEMS pressure sensor, some scholars have improved the sensor structure to enhance temperature stability. Cheng Pang and Xuzhi Chen et al. [32,33] designed a MEMS pressure sensor with a double F-P cavity structure, which can measure temperature and pressure simultaneously and calibrate the pressure with temperature. Wenhua Wang et al. [21] reduced the influence of thermal expansion coefficient, epoxy resin decomposition, and working point drift by improving the welding process and sensor structure, thus reducing the temperature sensitivity to 0.011 nm/°C. Vellaluru neeharika et al. [23] placed two waveguide gratings with the same temperature coefficients on the MEMS pressure sensor, eliminating the influence of temperature. Juncheng Xu et al. [34] retained part of the air in the F-P cavity during the fabrication of the sensor, which reduced the temperature dependence to 0.0076 psi/°C. Kumart pattnaik P. et al. [35] put two long-period gratings in the sensor, which reduced the temperature insensitivity. Tiegen Liu et al. [24] designed an optical fiber F-P pressure sensor for liquid level measurement, which adopts a silicon diaphragm and high borosilicate glass tube to eliminate the influence of residual gas temperature, and the measurement error caused by temperature drift is less than 0.09% F.S./K. Wenhua Wang et al. [27] designed an optical fiber F-P pressure sensor for liquid level measurement, which adopts an unsealed cavity structure to reduce the temperature dependence to 0.013 nm/°C. Zhiyuan Li et al. [36] proposed an intensity compensation demodulation method to eliminate the influence of temperature on the signal, reducing the intensity fluctuation of two orthogonal signals by 95.1% and 95.7%, respectively. Pinggang Jia et al. [37] proposed an F–P interferometric gas refractive-index sensor fabricated by inserting a single-mode fiber and a hollow silica tube. When the temperature changes, the thermal expansion of different parts of the sensor counteracts each other to achieve temperature compensation. Although the above method can reduce the influence of temperature on pressure measurement to a certain extent, it increases the complexity and puts forward higher requirements for the fabrication process.

In view of the above problems, this paper studies the temperature and pressure characteristics of the MEMS pressure sensor and proposes a temperature compensation method. In this method, the wavelength of the sensor at different temperatures is converted to the wavelength at a fixed temperature to eliminate the influence of temperature. Moreover, when calibrating the sensor, it is only necessary to calibrate the pressure-wavelength relationship at a certain temperature point so that the calibration operation is more operational.

## 2. Working Principle of MEMS Pressure Sensor

The structure diagram and the image of the MEMS pressure sensor are shown in Figure 1, which is mainly composed of the F-P cavity and grin-lens. The F-P cavity is composed of Silicon-On-Insulator (SOI) wafer and glass plate. The SOI wafer consists of top silicon, an intermediate oxide layer, and bottom silicon and is fixed on the glass plate by silicon glass anodic bonding. The side length of the F-P cavity is 3 mm, and the thickness is 0.9 mm. Since the F-P cavity is fragile and needs to be protected, the diameter of the pressure-sensing surface of the packaged sensor is 11 mm, and the length of the sensor is 500 mm. G.652 single-mode optical fiber (Manufacturer: Corning Optical Communications, Shanghai, China) is used in the sensor, and its model is 'Corning SMF-28'. The F-P cavity is made of silicon material with high-reflection film on both sides. After the light enters the F-P cavity, a multi-beam interference is formed between the two highly reflective films. Ignoring the power loss caused by the absorption and scattering of light by the two highly reflective films, the reflectivity of the F-P cavity can be expressed as [38]:

$$R_{F-P} = \frac{4R\sin^2\frac{\delta}{2}}{(1-R)^2 + 4R\sin^2\frac{\delta}{2}} \tag{1}$$

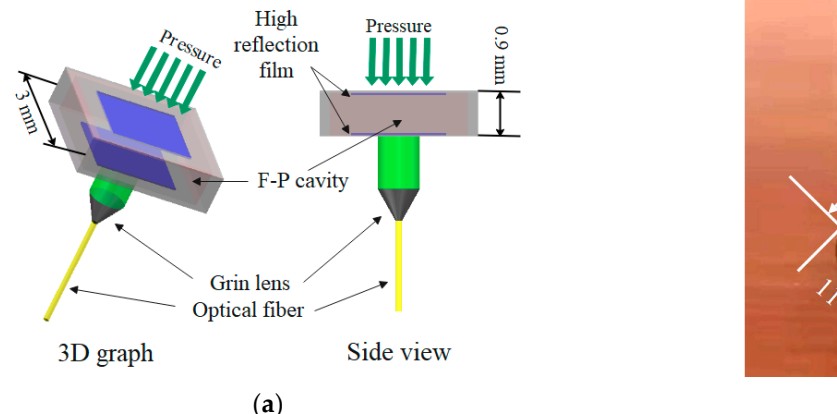
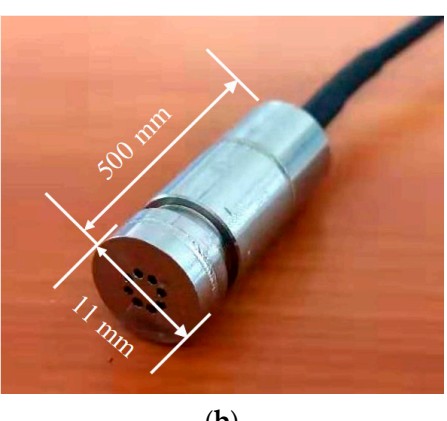

**Figure 1.** (**a**) Structure diagram and (**b**) image of the MEMS pressure sensor.

In Equation (1), $R$ is the reflectance of two high-reflection films, $\Delta$ is the phase shift formed by light returning once between two high-reflection films, and its value is [38]:

$$\delta = \frac{4\pi nL}{\lambda}. \tag{2}$$

In Equation (2), $n$ is the refractive index of the F-P cavity, $L$ is the cavity length, and $\lambda$ is the wavelength. From Equation (1), when $\delta/2 = m\pi$ ($m$ is a positive integer), the reflectivity is the minimum, and the following relationship can be further obtained.

$$2nL = m\lambda \tag{3}$$

It can be seen from Equation (3) that the wavelength is linearly related to the cavity length. When the pressure acting on the pressure sensing surface of the sensor changes, the length of the F-P cavity changes, which further leads to changes in wavelength. The grin lens connected to the F-P cavity converges the interference light into the optical fiber, and the demodulation device connected to it obtains the wavelength of the interference light, thus realizing pressure sensing.

## 3. Temperature and Pressure Characteristics of the Sensor

In the following, the temperature and pressure characteristics of the sensor are experimentally studied. The research method is to apply $i$ standard pressures and $j$ standard

temperatures to the sensor to obtain the wavelength $\lambda_{mn}(m = 1, 2, 3 \ldots i, n = 1, 2, 3 \ldots j)$, and study the change rule of $\lambda_{mn}$ with pressure and temperature.

Figure 2 shows the schematic diagram of the test system. The test system includes three parts: wavelength demodulator (Manufacturer: Shanghai Baiantek Sensing Technology Company, Shanghai, China. Model: BA-FT210H-M16), programmable temperature and humidity test chamber, and pressure source. The accuracy of the wavelength demodulator is ± 1 pm. In order to obtain high-precision test results, a high-precision pressure source composed of a gas tank, a pressure controller, and a vacuum pump is used, with an accuracy of 0.01% F.S. During the experiment, the pressure sensor is placed in the test chamber with temperature fluctuation less than 0.5 °C, and the gas pressure source is led into the test chamber to apply pressure to the sensor. The temperature in the test chamber is set to rise from 5 °C to 45 °C in steps of 5 °C. At each temperature point, the output of the pressure source is set to rise from 0 MPa to 4 MPa in steps of 1 MPa. The wavelength of the sensor at different temperatures and pressures can be obtained from the above tests, as shown in Figure 3.

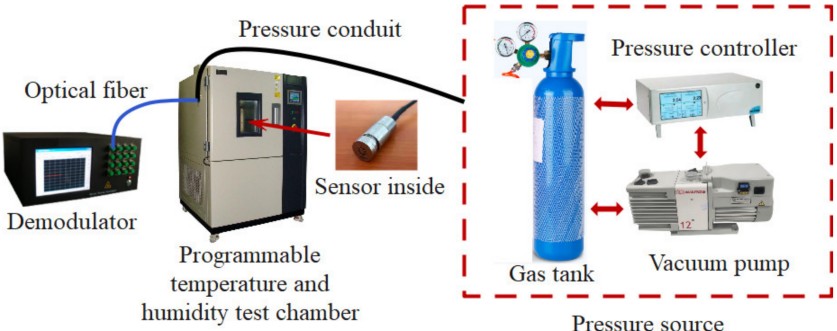

**Figure 2.** Schematic diagram of the test system.

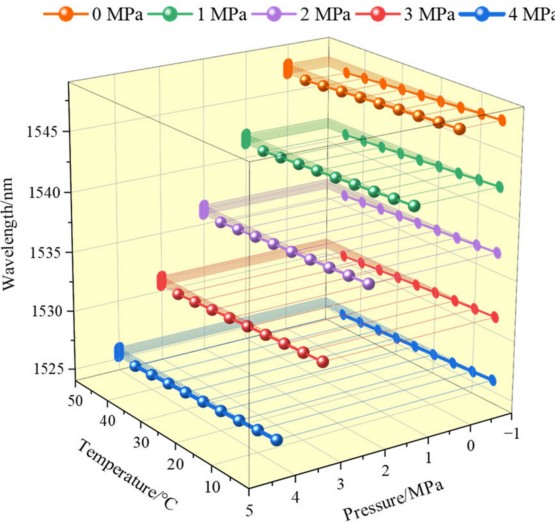

**Figure 3.** Variation of wavelength with temperature and pressure.

It can be seen from Figure 3 that at a fixed temperature, the wavelength decreases with the increase of pressure, and the pressure-wavelength relationship is approximately linear. When the temperature changes, the relative relationship between wavelength and pressure remains basically unchanged, but with the increase in temperature, the wavelength decreases under the same pressure.

To clearly illustrate the effect of temperature and pressure on wavelength, the data in Figure 3 are redrawn in two separate plots, as shown in Figure 4. Figure 4a shows the influence of temperature on wavelength under different pressures and lists the linear fitting equation of the temperature-wavelength relationship. It can be seen from Figure 4a that

the slope of the five fitting curves is not 0, which indicates that the temperature has an impact on the wavelength. At the same time, the slope of the five fitting curves is very similar, indicating that under different pressures, the changes in the wavelength caused by the same temperature change are almost identical. Figure 4b shows the influence of pressure on the wavelength at different temperatures, that is, the response of the sensor. Similar to Figure 4a, the slope of the fitting curve is very close, indicating that the influence of pressure on the wavelength is almost the same at different temperatures.

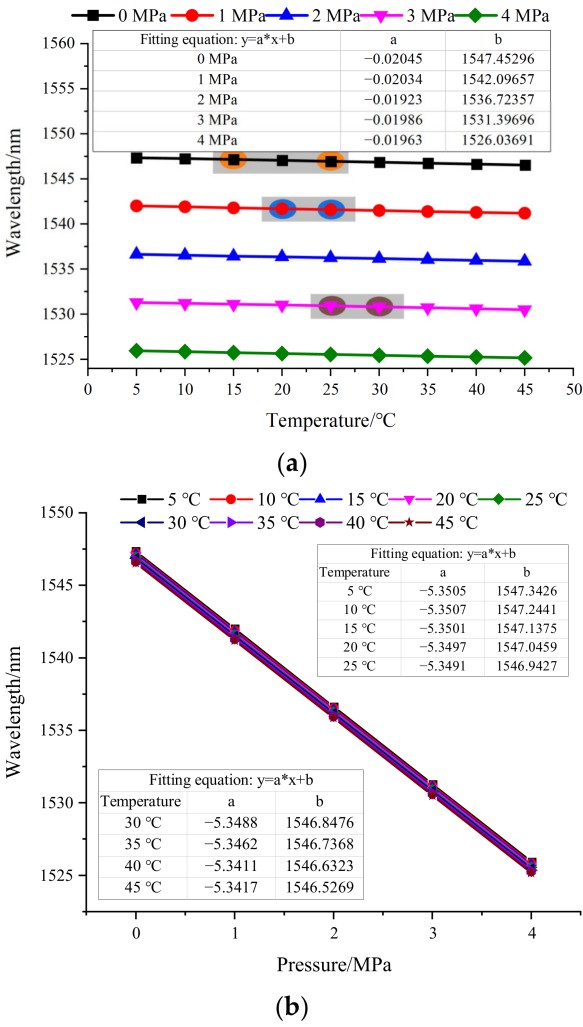

**Figure 4.** (**a**) Influence of temperature on wavelength under different pressure and (**b**) Influence of pressure on the wavelength at different temperatures.

When the pressure sensor is used for measurement, if the temperature effect is not compensated, the pressure is calculated using the same wavelength-pressure relationship in any temperature environment. As seen from Figure 3, when the temperature changes, the wavelength-pressure relationship will change, but in practical application, it is impossible to obtain the wavelength-pressure relationship at all temperature points, so it will bring error in calculating the pressure with the constant wavelength-pressure relationship. Next, the error of the data shown in Figure 3 is calculated using the wavelength-pressure relationship at 25 °C.

The pressure and wavelength data at 25 °C in Figure 4b are fitted to obtain the wavelength-pressure relationship. By substituting the wavelength measured at other temperatures into this relationship, the corresponding pressure value can be calculated. The test error is obtained by subtracting the pressure set by the gas pressure source from the calculated pressure value, which mainly reflects the influence of temperature on the

pressure sensor. The error curve is shown in Figure 5. As the fitting relationship at 25 °C is used to calculate the pressure under all temperature conditions, the error at 25 °C in Figure 5 is 0. The more the deviation from 25 °C, the greater the error, and the maximum error is −0.0796 MPa.

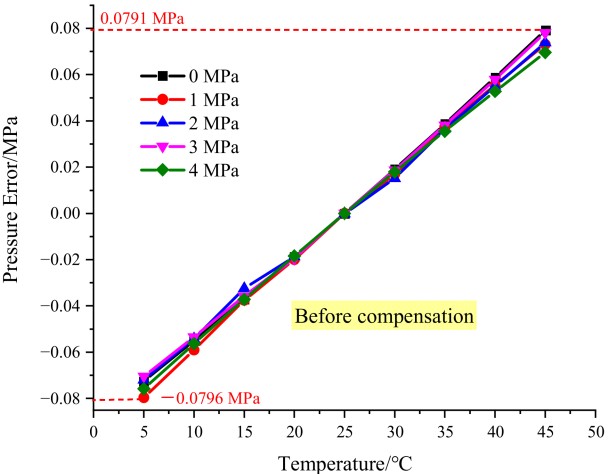

**Figure 5.** Test error before temperature compensation.

## 4. Temperature Compensation by Conversion Method

It can be seen from Figure 4 that under different temperatures, the same pressure changes correspond to basically the same wavelength changes, and under different pressures, the same temperature changes also correspond to basically the same wavelength changes. Therefore, the influence of temperature and pressure on the wavelength can be separated; that is, the wavelength of the sensor can be expressed as follows:

$$\lambda_{PT} = \lambda_{P_0 T_0} + \Delta\lambda_P + \Delta\lambda_T \tag{4}$$

In Equation (4), $\lambda_{P_0 T_0}$ represents the initial wavelength when the pressure is $P_0$ and the temperature is $T_0$. $\Delta\lambda_P$ and $\Delta\lambda_T$ represents the wavelength change caused by pressure and the wavelength change caused by temperature respectively.

When calculating pressure with wavelength, the influence of temperature can be eliminated by removing the part $\Delta\lambda_T$ in the wavelength change of Equation (4). $\Delta\lambda_T$ is the wavelength change relative to a certain temperature $T_0$, so the wavelength after removing this part can be expressed as:

$$\lambda_{PT_0} = \lambda_{PT} - \Delta\lambda_T \tag{5}$$

In Equation (5), $\lambda_{PT_0}$ represents the wavelength under the condition of $(P, T_0)$, which is equivalent to converting the wavelength under the condition of $(P, T)$ to the wavelength under the condition of $(P, T_0)$. Then the pressure can be calculated by using the pressure-wavelength relationship under the condition of $(P, T_0)$. The above methods are analyzed below.

It can be seen from Figure 4a that under a fixed pressure, the temperature-wavelength relationship is approximately linear. In order to make the temperature compensation more accurate, the quadratic relation is used to express the relationship between temperature and wavelength, as shown in Equation (6)

$$\Delta\lambda_T = a_0 + a_1(T - T_0) + a_2(T - T_0)^2 \tag{6}$$

In Equation (6), $a_0$, $a_1$ and $a_2$ are the temperature-wavelength compensation coefficients, which represent the temperature-wavelength relationship and can be calculated by the equations formed by the test data at different temperatures and pressures. For example, the wavelength is tested under the conditions of temperature $(T_0, T_1, T_2, T_3)$ and pressure $(P_1, P_1, P_3)$. Three groups of test data $((T_0, P_1)$ and $(T_1, P_1))$, $((T_0, P_2)$ and $(T_2, P_2))$, $((T_0, P_3)$

and $(T_3, P_3)$) are selected to form three equations for calculation. The equations composed of above three groups of data are shown in Equation (7), and $T_0$ is selected as the target temperature.

$$\begin{bmatrix} 1, (T_1 - T_0), (T_1 - T_0)^2 \\ 1, (T_2 - T_0), (T_2 - T_0)^2 \\ 1, (T_3 - T_0), (T_3 - T_0)^2 \end{bmatrix} \begin{bmatrix} a_0 \\ a_1 \\ a_2 \end{bmatrix} = \begin{bmatrix} \lambda_{P_1 T_1} - \lambda_{P_1 T_0} \\ \lambda_{P_2 T_2} - \lambda_{P_2 T_0} \\ \lambda_{P_3 T_3} - \lambda_{P_3 T_0} \end{bmatrix} \tag{7}$$

In Equation (7), $\lambda_{P_1 T_1}$ represents the wavelength under the condition of $(P_1,\ T_1)$, and other symbols have the same meaning.

After obtaining the temperature-wavelength compensation coefficients, the wavelength $\lambda_{PT0}$ at $T_0$ can be obtained by using the ambient temperature, the test wavelength, and Equations (4) and (5).

The relationship between the wavelength and the pressure can be expressed by Equation (8).

$$P_{T_0} = b_0 + b_1 \lambda_{\mathrm{PT}_0} + b_2 \lambda_{\mathrm{PT}_0}^2 + b_3 \lambda_{\mathrm{PT}_0}^3 \tag{8}$$

In Equation (8), $b_0, b_1, b_2$ and $b_3$ are the pressure-wavelength relationship coefficients when the temperature is $T_0$, which can be obtained by cubic fitting the test data. By substituting $\lambda_{PT_0}$ into Equation (8), the pressure $P_{T_0}$ at temperature $T_0$ can be calculated. Since the wavelength $\lambda_{PT_0}$ has been adjusted for temperature changes, the $P_{T_0}$ is the value that has eliminated the effect of temperature.

## 5. Verification of Compensation Method and Error Analysis

When processing data according to the above method, selecting different data points will result in different temperature-wavelength compensation coefficients $(a_0, a_1, a_2)$. When selecting different converted target temperatures, the pressure-wavelength relationship coefficients $(b_0, b_1, b_2, b_3)$ will also be different, so that the compensation effect may be affected. In order to verify the effectiveness of the method, in this part, different data points and converted target temperatures will be selected for temperature compensation of the test data shown in Figure 4.

### 5.1. Using 25 °C as the Converted Target Temperature for Temperature Compensation

In Figure 4a, the test data at temperatures of 15 °C, 20 °C, 25 °C, and 30 °C and pressures of 0 MPa, 1 MPa, and 3 MPa are selected and substituted into Equation (7) to form equations, as shown in Equation (9). Six test data are circled in Figure 4a, and two data with the same pressure form an equation. Solving Equation (9) can obtain the temperature-wavelength compensation coefficients $(a_0, a_1, a_2)$ when the converted target temperature is 25 °C.

$$\begin{bmatrix} 1, (15 - 25), (15 - 25)^2 \\ 1, (20 - 25), (20 - 25)^2 \\ 1, (30 - 25), (30 - 25)^2 \end{bmatrix} \begin{bmatrix} a_0 \\ a_1 \\ a_2 \end{bmatrix} = \begin{bmatrix} \lambda_{0\mathrm{MPa},15\,°C} - \lambda_{0\mathrm{MPa},25\,°C} \\ \lambda_{1\mathrm{MPa},20\,°C} - \lambda_{1\mathrm{MPa},25\,°C} \\ \lambda_{3\mathrm{MPa},30\,°C} - \lambda_{3\mathrm{MPa},25\,°C} \end{bmatrix} \tag{9}$$

After processing the data shown in Figure 4 according to the method in this paper, the compensated pressure value is calculated, and then the set value of the pressure source is subtracted to obtain the test error, as shown in Figure 6. The maximum error occurs at the temperature of 5 °C and the pressure of 1 MPa, and the maximum error is $-0.0152$ MPa. Compared with the error of $-0.0796$ MPa before temperature compensation, the error after compensation is significantly reduced.

### 5.2. Influence of Changing Converted Target Temperature on Temperature Compensation Effect

25 °C is used as the converted target temperature for the compensation processing of the data above. It can be seen from Figure 6 that the more the temperature deviates from 25 °C, the greater the test errors. If 5 °C or 45 °C is selected as $T_0$ for conversion, the errors when the test temperature deviates more from $T_0$ will become much larger after

temperature compensation. Next, the converted target temperature will be changed to 5 °C and 45 °C to test the effect of temperature compensation.

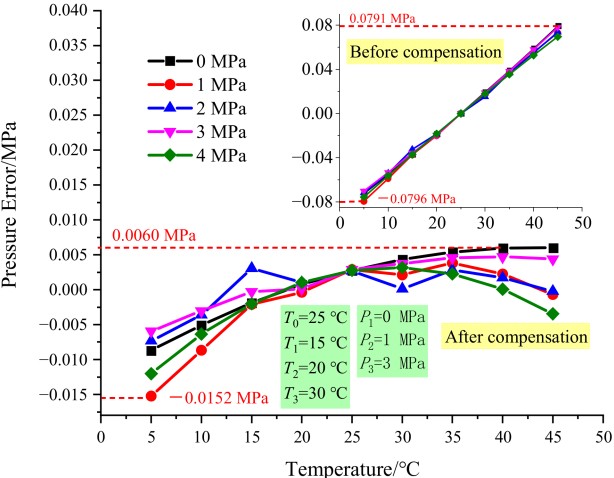

**Figure 6.** Errors when $T_0 = 25$ °C.

$T_1$, $T_2$, $T_3$, and $P_1$, $P_2$, $P_3$ in Equation (7) are consistent with that in Equation (9), $T_0$ is changed from 25 °C to 5 °C, and the test errors are recalculated. Figure 7 shows the test errors obtained before and after temperature compensation when the pressure-wavelength relation coefficients at 5 °C are used to calculate pressure. The maximum error before and after compensation is 0.1538 MPa and −0.0061 MPa, respectively. Similarly, the errors obtained are shown in Figure 8 when $T_0$ is changed to 45 °C. The maximum error before and after compensation is −0.1543 MPa and −0.0058 Mpa, respectively. It can be seen that the compensation algorithm is still effective when different converted target temperatures are selected.

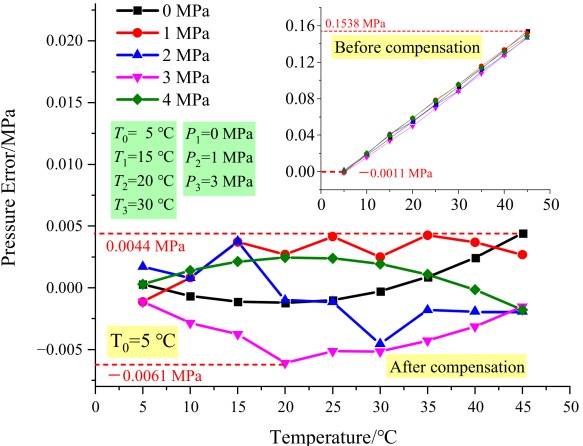

**Figure 7.** Errors when $T_0 = 5$ °C.

### 5.3. Influence of Changing Temperature and Pressure Data on Compensation Effect

As seen in Figure 4a, when the pressure is different, the influence of temperature on the wavelength is not the same. Therefore, when using Equation (7) to solve the temperature-wavelength compensation coefficients, different results will be obtained when selecting different temperature and pressure points in Figure 4a, resulting in different temperature compensation effects.

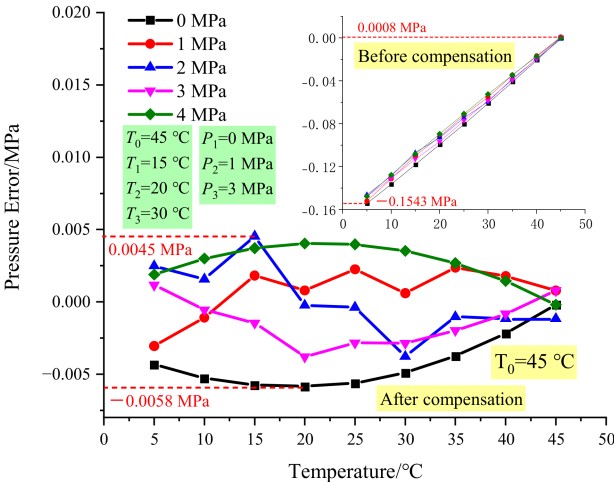

**Figure 8.** Errors when $T_0 = 45\,°C$.

In the following data processing, different data are selected and substituted into Equation (7) to verify the temperature compensation method. In order to make the results comparable, 25 °C is still selected as the target temperature, but the temperature and pressure data selected are changed.

First, the pressure data are kept the same as in Figure 6, which is 0 MPa, 1 MPa, and 3 MPa, and the temperature data are changed to 5 °C, 40 °C, and 45 °C. Figure 9a shows the results, and it shows that the maximum error is −0.0114 MPa. Then the temperature data are kept the same as in Figure 6, which is 15 °C, 20 °C, and 30 °C, and the pressure data are changed to 0 Mpa, 3 Mpa, and 4 MPa to compare the temperature compensation effect. Figure 9b shows the results, and the maximum error is −0.0167 MPa.

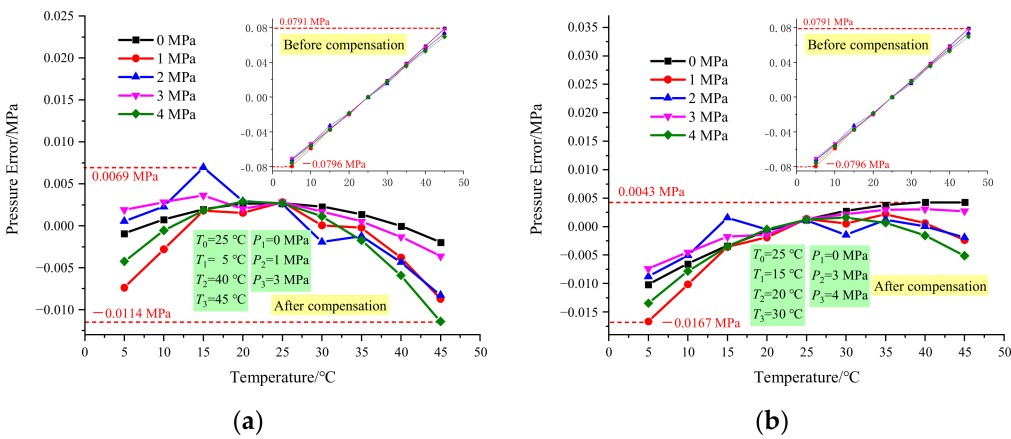

**Figure 9.** Errors with temperature compensation (**a**) the temperature is 5 °C, 40 °C and 45 °C and (**b**) the pressure is 0 MPa, 3 MPa and 4 MPa.

It can be seen that after changing the pressure and temperature data, compared with the error of 0.0796 MPa without temperature compensation, the temperature compensation effect is obvious.

The temperature-wavelength coefficients ($a_0$, $a_1$ and $a_2$) obtained from a group of data are used for the above temperature compensation. Next, the temperature wavelength coefficients averaged by multiple groups ($a_0$, $a_1$ and $a_2$) are used for temperature compensation to study the compensation effect. The converted target temperature in lines 2, 4, and 5 of Table 1 is 25 °C, so the corresponding ($a_0$, $a_1$ and $a_2$) of these three groups of data are selected for average processing, and then temperature compensation is calculated. The compensated results are shown in Figure 10. It can be seen from Figure 10 that when the averaged temperature-wavelength coefficients are used for temperature compensation, the

maximum error is −0.0131 MPa, and the accuracy is 0.33%. Table 1 shows that the errors obtained without averaging are −0.0152 MPa, −0.0114 MPa, and −0.0167 Mpa, respectively, and the corresponding accuracies are 0.38%, 0.29%, and 0.42%, respectively. It can be seen that when the averaged temperature-wavelength coefficients are used for compensation, the accuracy is better than the results of two of them and slightly worse than the highest accuracy, indicating that the averaged coefficients are better for temperature compensation.

**Table 1.** Errors comparison with different data.

| Temperature (°C) | | | Pressure (MPa) | | | Target Temperature (°C) | Before Compensation | | After Compensation | |
|---|---|---|---|---|---|---|---|---|---|---|
| | | | | | | | Errors (MPa) | Accuracy (F.S.) | Errors (MPa) | Accuracy (F.S.) |
| 15 | 20 | 30 | 0 | 1 | 3 | 5 | 0.1538 | 3.85% | 0.0044 | 0.11% |
| 15 | 20 | 30 | 0 | 1 | 3 | 25 | 0.0796 | 1.98% | −0.0152 | 0.38% |
| 15 | 20 | 30 | 0 | 1 | 3 | 45 | −0.1543 | 3.86% | −0.0058 | 0.15% |
| 5 | 40 | 45 | 0 | 1 | 3 | 25 | 0.0796 | 1.98% | −0.0114 | 0.29% |
| 15 | 20 | 30 | 0 | 3 | 4 | 25 | 0.0796 | 1.98% | −0.0167 | 0.42% |
| Using averaged temperature-wavelength coefficients | | | | | | 25 | 0.0796 | 1.98% | −0.0131 | 0.33% |

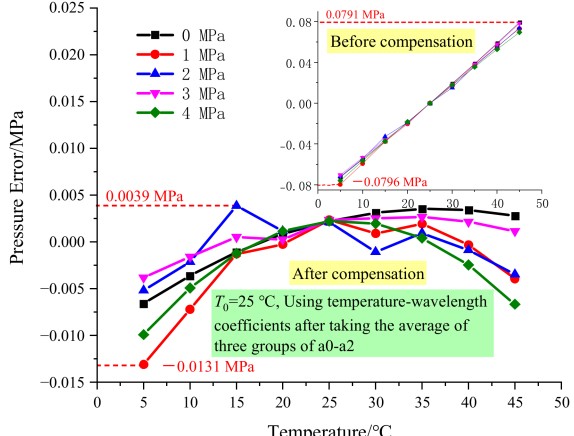

**Figure 10.** Errors with temperature compensation using averaged temperature-wavelength coefficients.

The temperature compensation method proposed in this paper is verified by changing the converted target temperature and data points. Table 1 summarizes the data points used in the method verification and the errors before and after compensation. It can be seen from Table 1 that when choosing different target temperatures and different data points, the errors will be affected to some extent. This is because the temperature-wavelength relationship and pressure-wavelength relationship of the sensor are not completely consistent under different temperature and pressure conditions. However, no matter how the data are selected, the compensation effect is very obvious.

## 6. Application of Temperature Compensation Method in Sensor Calibration

The temperature compensation method proposed in this paper can not only reduce the influence of temperature on pressure measurement but also be used in sensor calibration, which can make regular calibration more operable and greatly reduce the workload. The application of this method is described below.

The mechanical characteristics of the F-P cavity with MEMS structure will creep under long-term pressure, which will affect the accuracy of the pressure sensor. Next, the aging test of the sensors for two months is carried out to simulate the creep process and study the characteristics changes before and after creep. To facilitate comparison, the first test results before aging and the second test results after aging are drawn together, as shown in Figure 11.

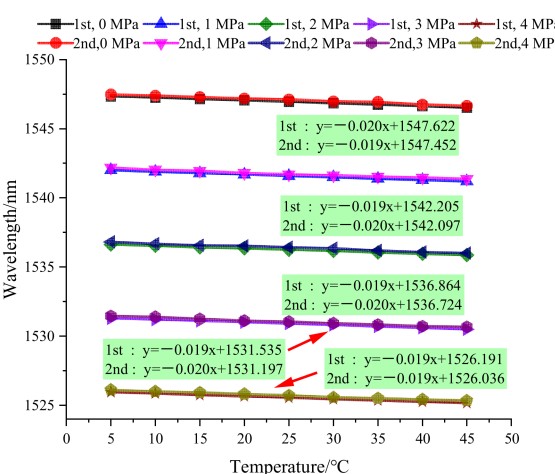

**Figure 11.** Sensor creep test.

Figure 11 shows that after two months of the aging test, the pressure-wavelength relationship has changed due to the creep; the wavelength becomes larger under the same pressure. It can be seen from the slope of each fitting curve that the influence of temperature on the wavelength change is basically unchanged; the wavelength change is 19–20 pm for a temperature change of 1 °C. This is because the mechanical characteristics of the sensor will change after aging, resulting in a change in the pressure-wavelength relationship, while the temperature characteristics are mainly determined by materials, so the temperature-wavelength relationship is basically unchanged.

After the creep of the sensor occurs, if the pressure is calculated with the coefficients obtained before the creep, it will bring great errors. The sensor needs to be calibrated to obtain new relationship coefficients. Normally, the calibration process is the process of recalibration. When using the method in Part 2 of this paper for calibration, if *m* temperature values and *n* pressure values are applied to the sensor, the data point to be tested is $m \times n$ ($9 \times 5$ in Figure 3), which takes a lot of time and imposes a great burden on sensor users.

When using the temperature compensation method in this paper for sensor calibration, only the pressure-wavelength relationship needs to be recalibrated at a certain temperature (such as 25 °C). The pressure-wavelength relationships at other temperature points do not need to be recalibrated and can be calculated using the method proposed in this paper, thus making the operation more tractable.

The data in Figure 11 are processed using the method described above. The pressure-wavelength coefficients obtained before and after the aging test are used to process the data. In order to verify the validity of the method, only the temperature-wavelength coefficients obtained before the aging test are used for the data processing. The test errors are shown in Figure 12. It should be noted that the black line '0 MPa' in Figure 12 represents the pressure applied during the test, while the '0 MPa' in the vertical axis represents the pressure error. It shows that the error obtained using the pressure-wavelength coefficients before the aging test is large, with a maximum value of −0.0430 MPa, and the errors are all less than 0 MPa, indicating that the sensor has drifted at all temperature points.

According to the calibration method proposed in this paper, the pressure-wavelength coefficients of the aged sensor are recalibrated at 25 °C with temperature compensation, and the error obtained is small, with a maximum value of 0.0137 MPa, and uniformly distributed above and below 0 MPa. It means that after aging, the pressure-wavelength relationship of the sensor has changed, while the temperature-wavelength relationship has changed very little. Therefore, by using the pressure-wavelength coefficients after calibration and the temperature-wavelength coefficients before calibration, high-accuracy measurement values can be obtained.

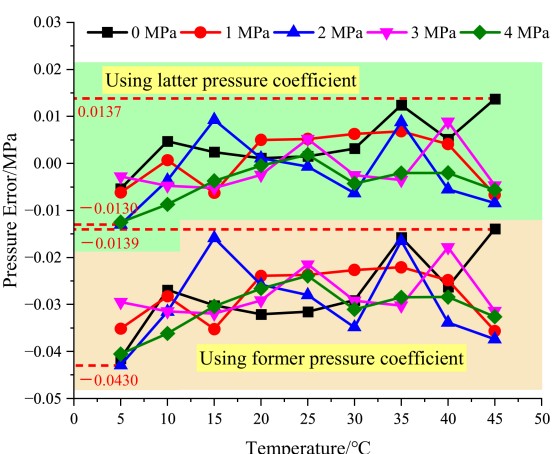

**Figure 12.** Pressure errors using the latter and former pressure-wavelength coefficients.

## 7. Discussion

The purpose of designing this sensor is to measure the depth of seawater, so only the sensor characteristics in the range of 5–45 °C are studied. The pressure characteristics of the SOI material used in the sensor are still very good in a high-temperature environment, and when the temperature increases to 300 °C, the sensor made with SOI still maintains good linearity [39–41]. However, when the temperature changes, the temperature drift characteristics of SOI materials are less studied. Therefore, in a high-temperature environment, the characteristics of the sensor still need to be further studied to confirm whether the methods described in this paper are effective.

Some F-P pressure sensor has large thermal drift, such as the PDMS (Polydimethylsiloxane)-based F-P cavities. PDMS is a kind of material with a high thermo optical coefficient and large elasticity, which can be filled in the F-P cavity as a medium or as an elastic film of the F-P cavity to realize temperature or pressure sensing. When PDMS is used as the medium of the F-P cavity, because the refractive index of PDMS is linear with temperature, it can be seen from Equation (3) that the wavelength of the F-P cavity is also linear with temperature. Therefore, in theory, the method in this paper can be used to eliminate the influence of temperature. If PDMS is used as the elastic film of the F-P cavity, its structure is similar to that of the sensor described in this paper, so it is also applicable to the temperature compensation method described in this paper. However, the above analysis is only from a theoretical point of view, without experimental verification. So, whether the sensors made of other materials are suitable for the temperature compensation proposed in this paper needs further research.

## 8. Conclusions

In this paper, the temperature characteristics of optical fiber MEMS pressure sensors are investigated, and a temperature compensation method by converting the wavelength is proposed. The tests show that when the converted temperature is 25 °C, the pressure measurement accuracy of the sensor is improved from 1.98% F.S. to 0.38% F.S. in the range of 5–45 °C and 0–4 MPa.

In addition to effectively eliminating the effect of temperature with this method, only the pressure-wavelength coefficients at a certain temperature point need to be corrected in the later sensor calibration without the need to calibrate over the full temperature range. The calibration results before and after sensor aging show that the method is effective and more operable.

**Author Contributions:** The work described in this article is the collaborative development of all authors. Conceptualization, Y.L.; methodology, Q.S. and H.F.; writing—original draft preparation, G.Y.; writing—review and editing, G.Y., Q.S. and H.F. All authors have read and agreed to the published version of the manuscript.

**Funding:** This research was funded by National Natural Science Foundation of China, grant number 61775057, Fundamental Research Funds for the Central Universities, grant number 2018MS097, Hebei Province Science and Technology Support Program, grant number SZX2020034.

**Institutional Review Board Statement:** Not applicable.

**Informed Consent Statement:** Not applicable.

**Data Availability Statement:** Data available on request from the authors. The data that support the findings of this study are available from the corresponding author, Guozhen Yao, upon reasonable request.

**Conflicts of Interest:** The authors declare no conflict of interest.

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
