# Peer review of "Research on Temperature Compensation of Optical Fiber MEMS Pressure Sensor Based on Conversion Method"

_photonics, doi:10.3390/photonics10010022_

Round 1

Reviewer 1 Report

This paper reported a study on temperature compensation for the optical fiber F-P pressure sensor. The cross-sensitivity of temperature and pressure is a common problem for the fiber F-P sensors. This work is meaningful for the sensing accuracy improvement of optical fiber F-P sensors. However, some details need to be addressed and the specific comments are given as follows.

1、         From Fig. 1, the dimension sensors probe is not in a MEMS scale. Where MEMS technology is used? This sensor is just a common extrinsic optical fiber F-P sensor.

2、         In the Fig. 3, the Y-axis, what does wavelength refer to? The wavelength of the peak or trough of interference spectrum? It is better to give a graph to display the response of the F-P sensor.

3、         For the Eq. (6), this is a general method for temperature sensitivity calibration of fiber F-P sensors. What’s the novelty of the method proposed by this paper?

4、         In the practical application for pressure measurement, if the ambient temperature is unknown, how to do? Does it mean that a temperature sensor is necessary?

Reviewer 2 Report

In this manuscript, the author demonstrates an optical fiber MEMS pressure sensor adopting the conversion method that can eliminate the influence from temperature variation. Due to the mutual-independence of the wavelength-changing effects from pressure and temperature, setting the latter to a fixed value, the temperature-related term can be subtracted to obtain a pressure-wavelength relation that is applicable under different temperatures, so that the error induced by temperature variation can be reduced and the complexity of calibration could be minished. I suggest the author to concern following issues:

1. To use a F-P cavity in front of a fiber tip for measuring pressure is not new. Related other similar geometires, I cannot see impressive enhancements in performance.

2. What kind of light source is used has not been mentioned in the manuscript, what’s its parameter such as power and stability.

3. Fig.2(b) is somewhat casual and messy, a neat system figure is needed to make it more concise and intuitive for the readers.

4. Why quadratic relation is adopted to express the relation between wavelength and temperature (Eq.(6)), while cubic relation is used to describe the relation between the wavelength and pressure(Eq.(8))?

5. The detail of demodulation is not given.

6. In line 349, ‘0 Mpa’ should be indicated to be the value of pressure error—the vertical axis in Fig.11, instead of the black line ‘0 Mpa’ in Fig.11, to avoid confusion.

7. To calculate the temperature-wavelength compensation coefficients a0-a2, only one group of test data((T0,P1), (T1,P1), (T0,P2), (T2,P2), (T0,P3) (T3,P3)) is used, how about calculating multiple sets of a0-a2 by feeding different groups of data and taking the average value, will this refine the precision?

8. Before the sensor is used, coefficients (a0-a2 and b0-b3) should be calculated first, can the same coefficients be applied to different devices with the same configuration to reduce the complexity of calculation?

Reviewer 3 Report

Reviewer’s Report

In this work, the authors proposed the temperature characteristics of optical fiber MEMS pressure sensors and a temperature compensation method by converting the wavelength.

This research is of great use in the sensor field. However, the necessary elucidation of the mechanism and references are absent in the manuscript to explain the obtained results. Therefore, in my opinion, a minor revision is needed to accommodate the high-quality requirements of this Journal. 

1.      Line 28, “F-P cavity” should be “Fabry–Perot cavity.” Please give a full name before abbreviating. In the same manner, please check the abbreviation throughout the manuscript.

2.      To show the novelty of this work and be beneficial for the readers to know the other types of optical pressure sensors, some references dealing with the optical pressure sensor and temperature sensor (J. Phys. D: Appl. Phys., 2020, 53 (11), 115401) should be included in the reference of introduction section.

3.      Line 96, the size of the F-P cavity (including the cavity length) and the proposed structure in Fig. 1 should be elucidated in more detail.

4.      Line 112, what kind of optical fiber you used should be described in the text.

5.      Line 139-141, the mechanism of “When the temperature changes, … but with the increase of temperature, the wavelength decreases under the same pressure.” should be clarified or quoted in the related literature.

6.      Please quote the related reference of Equations (1)-(9) if you referred to the corresponding articles.

Reviewer 4 Report

Recommendation: Major revisions are necessary

Comments:

This paper proposed a novel method for the temperature compensation of optical fiber MEMS pressure sensor. The author detailly investigated the theoretical analysis and experimental applications. This holds great value for future research in the field of F-P pressure sensors. To ensure the advantages, understandability, and logic of this paper, a major revision is required:

1)     The author investigated the behavior of the sensor under temperatures ranging from 5 to 45ºC and indicated the proposed compensation works well under this temperature range. However, many F-P pressure sensors are required to work at high-temperature. Does this method also work well under larger temperature ranges (such as 50-200 ºC)? The author should analyze this theoretically.

2)     The model of the silica F-P pressure sensor has a small thermal drift, according to Fig.4. However, some F-P pressure sensor has large thermal drift, such as the PDMS-based F-P cavities. Does this method also work well in this condition? The author should also analyze this.

3)     In the introduction, the citation of someone’s work should follow this: xxx et al…’. For instance, in line 45, the sentence should be written as ‘Jiang Xiaofeng et al designed an optical …’.

4)     In Figure 1b, a scale bar should be added. The following work on temperature compensation is important. The author should consider citing them:

Yixian Ge, et al, Temperature Compensation for Optical Fiber Graphene Micro-Pressure Sensor Using Genetic Wavelet Neural Networks, IEEE Sensors Journal, 2021, 21(21); Zhe Li, et al, Microbubble-based fiber-optic Fabry–Perot pressure sensor for high-temperature application, Applied Optics, 2018, 57(8); Pinggang Jia, et al, Temperature-compensated fiber-optic Fabry–Perot interferometric gas refractive-index sensor based on hollow silica tube for high-temperature application, Sensors and Actuators B: Chemical, 2017, 244.

5)     The language should be polished.

Round 2

Reviewer 2 Report

The authors generally well addressed my questions. Although I insist that a fiber F-P MEMS sensor is not new in physics (previous Q1), I admire that the authors can reinforce the technical details for readers who are interested in engineering operation. Therefore, I recommend to accept this manucript. 

Reviewer 4 Report

The authors have answered all my concerns. I recommend publishing it in this journal.